# Cadmium-Induced Proteinuria: Mechanistic Insights from Dose–Effect Analyses

**DOI:** 10.3390/ijms24031893

**Published:** 2023-01-18

**Authors:** Soisungwan Satarug, David A. Vesey, Glenda C. Gobe

**Affiliations:** 1Kidney Disease Research Collaborative, Translational Research Institute, Brisbane, QLD 4102, Australia; 2Department of Nephrology, Princess Alexandra Hospital, Brisbane, QLD 4102, Australia; 3School of Biomedical Sciences, The University of Queensland, Brisbane, QLD 4102, Australia; 4NHMRC Centre of Research Excellence for CKD QLD, UQ Health Sciences, Royal Brisbane and Women’s Hospital, Brisbane, QLD 4102, Australia

**Keywords:** albumin, albumin-to-creatinine ratio, α1-microglobulin, β_2_-microglobulin, cadmium, creatinine clearance, estimated glomerular filtration rate, protein reabsorption, tubulopathy, urine total protein

## Abstract

Cadmium (Cd) is a toxic metal that accumulates in kidneys, especially in the proximal tubular epithelial cells, where virtually all proteins in the glomerular ultrafiltrate are reabsorbed. Here, we analyzed archived data on the estimated glomerular filtration rate (eGFR) and excretion rates of Cd (E_Cd_), total protein (E_Prot_), albumin (E_alb_), β_2_-microglobulin (E_β2M_), and α1-microglobulin (E_α1M_), which were recorded for residents of a Cd contamination area and a low-exposure control area of Thailand. Excretion of Cd and all proteins were normalized to creatinine clearance (C_cr_) as E_Cd_/C_cr_ and E_Prot_/C_cr_ to correct for differences among subjects in the number of surviving nephrons. Low eGFR was defined as eGFR ≤ 60 mL/min/1.73 m^2^, while proteinuria was indicted by E_Pro_/C_cr_ ≥ 20 mg/L of filtrate. E_Prot_/C_cr_ varied directly with E_Cd_/C_cr_ (β = 0.263, *p* < 0.001) and age (β = 0.252, *p* < 0.001). In contrast, eGFR values were inversely associated with E_Cd_/C_cr_ (β = −0.266, *p* < 0.001) and age (β = −0.558, *p* < 0.001). At E_Cd_/C_cr_ > 8.28 ng/L of filtrate, the prevalence odds ratios for proteinuria and low eGFR were increased 4.6- and 5.1-fold, respectively (*p* < 0.001 for both parameters). Thus, the eGFR and tubular protein retrieval were both simultaneously diminished by Cd exposure. Of interest, E_Cd_/C_cr_ was more closely correlated with E_Prot_/C_cr_ (*r* = 0.507), E_β2M_ (*r* = 0.430), and E_α1M_/C_cr_ (*r* = 0.364) than with E_Alb_/C_cr_ (*r* = 0.152). These data suggest that Cd may differentially reduce the ability of tubular epithelial cells to reclaim proteins, resulting in preferential reabsorption of albumin.

## 1. Introduction

Cadmium (Cd) is an environmental contaminant of continuing public health concern worldwide because it is detectable in most food types; as such, diet forms the main source of exposure in non-occupationally exposed and non-smoking populations [1,2]. Multiple organ systems, including kidneys [1,2], bone [3], liver [4,5] and the central nervous system [6], are susceptible to the toxicity of Cd, even at low body burdens. The cytotoxicity of Cd has been demonstrated in nearly all cell types, such as erythrocytes and the tubular epithelial cells of kidneys, which are known to actively accumulate Cd [1,2,7].

The pivotal role played by kidney tubular epithelial cells in reuptake and excretion of proteins is gaining support from recent research data [8,9,10,11,12,13]. An approximate 40–50 g of protein may reach the urinary space daily, and virtually all of it is reabsorbed [8,9,10,11,12,13]. The majority of protein in the glomerular ultrafiltrate is retrieved in the S1 sub-segment of the proximal tubule, where the receptor-mediated endocytosis involving the megalin/cubillin system is involved [8,9]. Protein reabsorption occurs also in the distal tubule and the collecting duct, where the process is mediated by the neutrophil gelatinase-associated lipocalin (NGAL)/lipocalin-2 receptor system [14,15,16].

The protein albumin, with a molecular weight of 66 kDa, is synthesized in the liver and ordinarily secreted into the circulation at a rate of 10–15 g per day [12,13]. Catabolism in muscle, the liver, and the kidney proximal tubular epithelial cells balance synthesis, and homeostasis is continued. In good health, the plasma concentration of albumin is between 3.5 g/dL and 5 g/dL, and the average half-life in plasma is 19 days [12,13]. Albumin is not normally filtered by glomeruli, due to its large molecular weight and its negative charge. By means of transcytosis through endothelial cells and podocyte foot processes, albumin enters the urinary space at a rate of 1–10 g per day [8,9,10,11,12,13,17].

Albuminuria is diagnosed when excretion of albumin, measured as albumin-to-creatinine ratio, rises to levels above 20 and 30 mg/g creatinine in men and women, respectively [18,19,20]. The persistence of albuminuria for at least three months is a diagnostic criterion of CKD. A progressive decrease in the eGFR below 60 mL/min/1.73 m^2^, termed low eGFR, is also a diagnostic criterion of CKD [18,19,20].

An elevated excretion of low-molecular-weight proteins, namely, retinol binding protein, β_2_-microglobulin (β_2_M), and α1-microglobulin (α_1_M), has been the most frequently investigated [1,2]. The protein β_2_M, with a molecular weight of 11,800 Da, is synthesized and shed by all nucleated cells in the body [21]. By virtue of its small mass, β_2_M is filtered freely by the glomeruli and is reabsorbed almost completely by the kidney’s tubular cells [22]. Cd has been shown to cause a reduction in the tubular maximum reabsorption of β_2_M [23], and increased β_2_M excretion has been used as an indicator of tubulopathy for many decades. However, our previous study showed that β_2_M excretion of 100–299, 300–999, and ≥ 1000 μg/g creatinine was associated with 4.7-, 6.2-, and 10.5-fold increases in the risk of an estimated glomerular filtration rate (eGFR) ≤ 60 mL/min/1.73 m^2^, which is commensurate with CKD [24]. These data suggest that an increased excretion of β_2_M above 300 µg/day, termed β_2_-microglobunuria, could be a consequence of Cd-induced tubulopathy in conjunction with nephron loss, evident from a reduction in the eGFR to 60 mL/min/1.73 m^2^ or below [24].

The present study aimed to evaluate the concurrent effects of Cd accumulation in kidneys on GFR reduction and excretion rates of proteins of high and low molecular weights, namely, total protein, albumin, β_2_M, and α_1_M. To enable an accurate assessment of these effects of Cd, we normalized excretion of Cd and all excreted proteins to creatinine clearance (C_cr_). This C_cr_-normalization corrects for differences in the number of surviving nephrons among study subjects, and it depicts an amount of a given chemical excreted per volume of filtrate, which is at least roughly related to the amount of the chemical excreted per nephron [25].

## 2. Results

### 2.1. Descriptive Characteristics of Study Subjects According to eGFR

Among 405 cohort participants, 190 (46.9%) were residents of a low-exposure area (Bangkok), and 215 (53.1%) persons lived in a high-exposure area of Mae Sot District, where Cd pollution was endemic (Table 1).

The overall mean age was 44.6 years. Of the total participants, 45.9% were smokers, including those who had quit less than 10 years ago. The percentages of females, hypertension, and diabetes were 51.4%, 13.8%, and 2.7%, respectively. The overall % of subjects with proteinuria, defined as urine protein ≥ 20 mg/L was 28.6%. The % of proteinuria was 30.6% when protein excretion (E_Prot_) was normalized to C_cr_ and E_Prot_/C_cr_ × 100 ≥ 20 mg/L was a cutoff value.

By eGFR stratification, 207 (51.1%), 147 (36.3%), and 51 (12.6%) had eGFR values > 90, 61–90, and ≤60, mL/min/1.73 m^2^, respectively. For simplicity, the eGFR values > 90, 61–90, and ≤60 mL/min/1.73 m^2^ were referred to as high, moderate, and low, respectively. The distributions of men and women across these three eGFR groups were similar. The low-eGFR group was the oldest, with the highest % of smoking, diabetes, and hypertension.

The % of proteinuria (urine protein ≥ 20 mg/L) in the low-, moderate-, and high-eGFR groups were 80.4%, 45.6%, and 3.9%, respectively (*p* < 0.001). The corresponding percentages of proteinuria were 86.3%, 49%, and 3.9% when (E_Prot_/C_cr_) × 100 ≥ 20 mg/L filtrate was defined as proteinuria.

Mean plasma creatinine, mean urine creatinine, mean urine protein, and mean urine Cd concentrations all were highest, middle, and lowest in the low-, moderate-, and high-eGFR groups, respectively (*p* < 0.001). Mean plasma total protein and mean plasma albumin in the high- and moderate-eGFR groups did not differ. Mean values of E_Cd_/E_cr_ and E_Prot_/E_cr_ were highest, middle, and lowest in the low-, moderate-, and high-eGFR groups, as were mean values of E_Cd_/C_cr_ and E_Prot_/C_cr_ (*p* < 0.001).

### 2.2. Predictors of Protein Excretion

Table 2 provides results of the multiple regression modeling of protein excretion as log[(E_Prot_/C_cr_ × 10^5^) where Cd excretion was incorporated as log[(E_Cd_/Ccr × 10^5^) along with other five independent variables (age, diabetes, sex, hypertension, and smoking).

Age, E_Cd_/C_cr_, diabetes, sex, hypertension, and smoking together accounted for 30.6%, 37%, and 25.9% of total variation in E_Pro_t/C_cr_ among all subjects (*p* < 0.001), men (*p* < 0.001), and women (*p* < 0.001), respectively. Age and E_Cd_/C_cr_ were independently associated with E_Prot_/C_cr_ in men and women. A positive association of E_Prot_/C_cr_ and E_Cd_/C_cr_ was stronger in men (β = 0.376, *p* < 0.001) than women (β = 0.179, *p* = 0.050).

An additional regression analysis of E_Prot_/C_cr_ was undertaken to assess a potential influence of plasma protein/albumin levels using data from 190 subjects of the low-exposure group. Plasma protein, plasma albumin, age, log [(E_Cd_/C_cr_) × 10^5^], sex, and smoking altogether did not explain the E_Prot_/C_cr_ variability (adjusted R^2^ = 0.008, *p* = 0.281). Associations of E_Prot_/C_cr_ with plasma protein, plasma albumin, and other independent variables were all not significant (*p* > 0.05).

### 2.3. Predictors of eGFR Deterioration

Table 3 provides results of multiple regression models of eGFR in which Cd excretion as log[(E_Cd_/Ccr × 10^5^) was incorporated together with five over independent variables (age, diabetes, sex, hypertension, and smoking).

Age, E_Cd_/C_cr_, diabetes, sex, hypertension, and smoking together accounted for 65%, 68%, and 63.5% of total variation in eGFR among all subjects (*p* < 0.001), men (*p* < 0.001), and women (*p* < 0.001), respectively. Lower eGFR values were associated with older age (β = −0.558, *p* < 0.001) and higher E_Cd_/C_cr_ (β = −0.266, *p* < 0.001). An inverse relationship between eGFR and E_Cd_/C_cr_ Cd excretion rates in women (β = −0.334, *p* < 0.001) was stronger than men (β = −0.178, *p* = 0.012).

### 2.4. Prevalence Odds Ratios for Proteinuria and Low eGFR in Relation to Cadmium Exposure

Table 4 provides results of logistic regression modeling where the prevalence odds ratios (POR) for proteinuria and low eGFR were determined for subjects of three Cd exposure groups. The independent variables included age, diabetes, sex, hypertension, and smoking.

The POR for proteinuria was lower at younger ages (POR = 0.923, *p* < 0.001), as was the POR for low eGFR (POR = 0.888, *p* < 0.001). The POR for proteinuria was increased 4.6-fold in those with (E_Cd_/C_cr_) × 100 > 8.28 µg/L filtrate (*p* < 0.001). Normotension was associated with a 64% decrease in POR for low eGFR (*p* = 0.016). The effect of hypertension on the POR for proteinuria was insignificant (*p* = 0.055). A dose–effect relationship was seen between POR for low eGFR and E_Cd_/C_cr_; the POR for low eGFR was increased 4.6-fold in those with (E_Cd_/C_cr_) × 100 ranging between 2.72 and 8.28 µg/L filtrate (*p* = 0.035), and the POR for low eGFR rose to 5.1 at E_Cd_/C_cr_) × 100 > 8.28 µg/L filtrate (*p* < 0.001).

### 2.5. Excretion Rates of Various Proteins and Cadmum in the High-Exposure Group

We analyzed the urine composition of 215 subjects who resided in a Cd-contaminated area in an attempt to quantify the influence of the number of surviving nephrons and Cd exposure. By eGFR stratification, there were 33 (15.3%), 131 (61%), and 51 (23.7%) who had eGFR > 90, 61–90, and ≤60 mL/min/1.73 m^2^, respectively (Table 5).

Relative to the high-eGFR group, the excretion of creatinine tended to rise in the moderate- and the low-eGFR groups (*p* = 0.054), while the urinary Cd concentrations (µg/L) in three eGFR groups showed no variation (*p* = 0.646) (Table 5). Consequently, the Cd body burdens of subjects in the three eGFR groups were not distinguishable (*p* = 0.079).

Distinctively, differences in body burden of Cd were apparent when E_Cd_ was normalized to C_cr_; the mean E_Cd_/C_cr_ was highest, middle, and lowest in the low-, moderate-, and high-eGFR groups (*p* < 0.001). As expected, those in the low-eGFR group excreted β_2_M, α_1_M, albumin, total protein, and Cd at the highest rates.

The % of E_β2M_/E_cr_ ≥ 1000 µg/g creatinine were 25.6%, and more than half (66.7%) of those with such a high E_β2M_ excretion were in the low eGFR group. The % of E_β2M_/E_cr_ ≥ 1000 µg/g creatinine was lowest, middle, and highest in the high-, moderate-, and low-eGFR groups, respectively (*p* < 0.001).

Only one subject (0.5%) had E_Alb_/E_cr_ > 300 mg/g creatinine. E_Alb_/E_cr_ values in the majority of subjects (84.2%) were < 30 mg/g creatinine. Only 15.3% had microalbuminuria (E_Alb_/E_cr_ values of 30–300 mg/g creatinine). The % of those with microalbuminuria was lowest, middle, and highest in the high-, moderate-, and low-eGFR groups, respectively (*p* < 0.001).

### 2.6. Correlation Analysis of Age, BMI, and Chemical Constiuents of Urine

We next undertook a correlation analysis of nine parameters that included age, BMI, excretion of β_2_M, α_1_M, albumin, total protein, and Cd (Table 6).

With the exception of BMI, age correlated positively with all other variables. BMI was negatively correlated with age, E_β2M_/C_cr_, and E_α1M_/_cr_. The excretion of creatinine correlated positively with age, E_Cd_/C_cr_, and E_Prot_/C_cr_. This parameter did not correlate with E_Alb_/C_cr_, E_β2M_, or E_α1M_/C_cr_. E_Cd_/C_cr_ was more closely correlated with E_Prot_/C_cr_ (r = 0.507), E_β2M_ (r = 0.430), and E_α1M_/C_cr_ (r = 0.364) than with E_Alb_/C_cr_.

Among three proteins quantified, β_2_M and α_1_M represented low-molecular-weight proteins. Albumin represented a high-molecular-weight protein, while urinary total protein was a measure of all proteins. As expected, E_Pro_/C_cr_ was highly correlated with all three proteins, β_2_M, α_1_M, and albumin (r = 0.508 − 0.546).

Figure 1 and Figure 2 provide scatterplots showing differential effects of Cd exposure on protein excretion in subsets of subjects stratified by nephron mass, based on eGFR > 60 and ≤60 mL/min/1.73 m^2^.

Stratification of subjects by eGFR uncovered differences in renal handling of albumin vs. β_2_M and α_1_M in response to Cd. Higher E_β2M_ and E_α1M_ values were associated with higher E_Cd_ in both eGFR groups (Figure 1a,b). The associations of E_β2M_, E_α1M_, and E_Cd_ all were stronger in the low-eGFR group than the high-eGFR group. In comparison, the relationship between E_Alb_ and E_Cd_ was insignificant in both eGFR groups (Figure 2a). An association between E_Alb_ and E_Prot_ was particularly strong in the low-eGFR group (Figure 2b), as were the associations of E_Alb_ and E_α1M_ (Figure 2d) and of E_Prot_ and E_α1M_ (Figure 1d). In comparison, the relationships between E_Alb_ and E_β2M_ were similar in the two eGFR groups (Figure 2c), as were the relationships between E_Prot_ and E_β2M_ (Figure 1c).

## 3. Discussion

A defective tubular reabsorption of the low-molecular-weight filterable protein β_2_M, referred to as tubular proteinuria, has been the most frequently reported adverse effect of environmental exposure to Cd. Analytical and biochemical epidemiologic research dissecting Cd-induced proteinuria is lacking. As demonstrated in the present study, mechanistic insights into proteinuria following long-term exposure to Cd and its accumulation in tubular epithelial cells of kidneys emerged when excretion rates of various proteins and Cd were normalized to creatinine (C_cr_). C_cr_-normalization corrects for differences among subjects in number of surviving nephrons. A conventional method of normalization of the rate of excretion of a given substance to creatinine excretion (E_cr_) corrects for urine dilution, but it introduces high variance into the dataset, because E_cr_ is not related to Cd exposure nor the function of kidneys. This has led to erroneous data interpretations in the past.

As shown in Table 5, the exposure levels of Cd, measured as E_Cd_/E_cr_ among groups of residents of a high-exposure area, did not differ statistically (*p* = 0.641); subjects in all three eGFR groups excreted a similarly high E_Cd_/E_cr_ of 10 µg/g creatinine. As a result of the indistinguishable body burden, a dose–effect relationship between excretion of various proteins and a Cd exposure measure could not be established.

In comparison, Cd exposure, measured as E_Cd_/C_cr_, increased from 8.1 to 9.7 and then 17 µg/L of filtrate in subjects with high, moderate, and low eGFR, respectively (*p* < 0.001). Excretion rates of total protein, Alb, β_2_M, and α_1_M all increased in parallel to E_Cd_/C_cr_ increment. In effect, a clear dose–response between E_Cd_/C_cr_ and excretion rates of total protein, Alb, β_2_M, and α_1_M was evident.

A further analysis of the relationship of the exposure measure (E_Cd_/C_cr_) and excretion rates of total protein, Alb, β_2_M, and α_1_M, shown in Figure 1 and Figure 2, provided further evidence for preferential reabsorption of albumin. An association between E_Alb_ and E_Cd_ was absent in both low- and high-eGFR groups (Figure 2a). This finding may be interpreted to suggest that most Alb is reabsorbed, consistent with a current view that albumin is reabsorbed almost completely and that a fraction of it is returned to the systemic circulation [10,11,12,13,14,15].

In an experimental study, Cd was found to disable the cubilin/megalin receptor system of albumin endocytosis, leading to albuminuria [26]. In addition, Cd diminished expression of megalin and ClC5 channels [27]. Cd may also increase glomerular permeability to albumin, as shown in another study, where a non-cytotoxic concentration of Cd (1 µM) increased the permeability of human renal glomerular endothelial cells in monolayers and caused the redistribution of the adherens junction proteins vascular endothelial-cadherin and β-catenin [28,29].

Of concern, CKD is a progressive syndrome with high morbidity and mortality and affects 8% to 16% of the world’s population, with increasing incidence worldwide [30,31,32,33]. Here, we have shown that Cd simultaneously increased risks of low eGFR and proteinuria 4- to 5-fold (Table 4). Indeed, associations between Cd exposure and low eGFR and albuminuria were repeatedly observed among participants in the U.S. National Health and Nutrition Examination Survey (NHANES) undertaken between 1999 and 2016 [34,35,36,37]. Increased E_alb_ was associated with a urinary Cd level as low as 0.22 μg/L in a study of participants in NHANES 2009–2012, aged ≥20 years [38], while an increase in risk of CKD among NHANES 1999–2006 was associated with E_Cd_/E_cr_ values ≥ 1 µg/g creatinine [34]. In a Spanish population study, a urinary Cd excretion (E_Cd_) of 0.27 µg/g creatinine was associated with an increase in the risk of albuminuria by 58% [39]. In another study, E_Cd_ > 1.72 µg/g creatinine was associated with elevated E_alb_ in Shanghai residents [40]. In a systematic and meta-analysis of pooled data from 28 studies, Cd exposure was found to increase the risk of proteinuria by 48% [41]. The pathogenesis of Cd-induced proteinuria, especially in low environmental Cd exposure conditions, requires further study, especially in experimentation, given that the glomerular and tubular causes of albuminuria may not be distinguishable in epidemiologic studies.

The results of the present study have strongly implicated Cd exposure as a risk factor for CKD. Minimization/avoidance of Cd exposure from smoking and habitual consumption of foods containing high levels of the metal are warranted. Kidney fibrosis after chronic exposure to Cd has been demonstratable in experimental studies [42,43]. Evidence from the synchrotron imaging of metals in human kidney tissue samples is in line with Cd-induced fibrosis [44]. The degree of tubular atrophy was positively associated with the level of Cd accumulation in a histopathological examination of kidney biopsies from healthy kidney transplant donors [45]. A prospective cohort study of Japanese residents in an area similarly polluted by Cd reported a 49% increase in mortality from kidney failure among women after adjustment for potential confounding factors [46].

## 4. Materials and Methods

### 4.1. Participants

We assembled data from 405 persons (197 men and 208 women) who participated in the larger population-based studies undertaken in a low-exposure area (Bangkok), and an endemic area of Cd contamination in the Mae Sot District, Tak Province, Thailand [47]. The Institutional Ethical Committee, Chiang Mai University and the Mae Sot Hospital Ethical Committee approved the study protocol. At the time of recruitment, all participants had lived at their current addresses for at least 30 years, and all gave informed consent prior to their participation. Exclusion criteria were pregnancy, breastfeeding, a history of metal work, and a hospital record or physician’s diagnosis of an advanced chronic disease. Smoking, diabetes, hypertension, regular use of medications, educational level, occupation, and family health history were ascertained by questionnaire. Diabetes was defined as fasting plasma glucose levels ≥ 126 mg/dL or a physician’s prescription of anti-diabetic medications. Hypertension was defined as systolic blood pressure ≥ 140 mmHg, diastolic blood pressure ≥ 90 mmHg, a physician’s diagnosis, or prescription of anti-hypertensive medications.

### 4.2. Collection and Analysis of Blood and Urine Samples

Simultaneous blood and urine sampling is required to normalize E_Cd_ to C_cr_. Accordingly, second morning urine samples were collected after an overnight fast, and whole blood samples were obtained within 3 h after the urine sampling. Aliquots of urine, whole blood, and plasma were stored at −20 °C or −80 °C for later analysis. The assay for urine and plasma concentrations of creatinine ([cr]_u_ and [cr]_p_) was based on the Jaffe reaction. The assay of β_2_M in urine ([β_2_M]_u_) was based on the latex immunoagglutination method (LX test, Eiken 2MGII; Eiken and Shionogi Co., Tokyo, Japan). The assay of urinary albumin ([alb]_u_) was based on a turbidimetric method (UniCel^®^ DxC800 Synchron system, Beckman Coulter, Fullerton, CA, USA).

For the Bangkok group, the urinary concentration of Cd ([Cd]_u_) was determined by inductively coupled plasma mass spectrometry (ICP/MS, Agilent 7500, Agilent Technologies, Santa Clara, CA, USA) because it had the high sensitivity required to measure very low Cd concentrations. Multi-element standards (EM Science, EM Industries, Inc., Newark, NJ, USA) were used to calibrate the Cd analyses. The accuracy and precision of those analyses were ascertained with reference urine (Lyphochek^®^, Bio-Rad, Sydney, Australia). The low limit of detection (LOD) of urine Cd, calculated as 3 times the standard deviation of blank measurements, was 0.05 µg/L. The Cd concentration assigned to samples with Cd below the detection limit was the detection limit divided by the square root of 2 [48].

For the Mae Sot group, [Cd]_u_ was determined with atomic absorption spectrophotometry (Shimadzu Model AA-6300, Kyoto, Japan). Urine standard reference material No. 2670 (National Institute of Standards, Washington, DC, USA) was used for quality assurance and control purposes. The low limit of detection of Cd quantitation, defined as 3 times the standard deviation of blank measurements, was 0.06 µg/L. None of the urine samples contained [Cd]_u_ below the detection limit.

### 4.3. Estimated Glomerular Filtration Rates and CKD Stratified by the KDIGO Categories

The GFR is the product of nephron number and mean single nephron GFR, and, in theory, GFR is indicative of nephron function [19,20,49]. In practice, the GFR is estimated from established Chronic Kidney Disease Epidemiology Collaboration (CKD-EPI) equations and is reported as eGFR [20,49].

Male eGFR = 141 × [plasma creatinine/0.9]^Y^ × 0.993^age^, where Y = −0.411 if [cr]_p_ ≤ 0.9 mg/dL and Y = −1.209 if [cr]_p_ > 0.9 mg/dL. Female eGFR = 144 × [plasma creatinine/0.7]^Y^ × 0.993^age^, where Y = −0.329 if [cr]_p_ ≤ 0.7 mg/dL and Y = −1.209 if [cr]_p_ > 0.7 mg/dL.

For dichotomous comparisons, CKD was defined as eGFR ≤ 60 mL/min/1.73 m^2^ or ACR > 30 mg/g creatinine. The KDIGO categories of CKD stages 1, 2, 3a, 3b, 4, and 5 corresponded to eGFR of 90–119, 60–89, 45–59, 30–44, 15–29, and <15 mL/min/1.73 m^2^, respectively. The KDIGO classification of ACR < 30, 30–300, and >300 mg/g creatinine corresponded to normal albumin excretion, microalbuminuria, and a severely elevated albumin excretion [19].

### 4.4. Normalization of ECd to Ecr and Ccr

E_x_ was normalized to E_cr_ as [x]_u_/[cr]_u_, where x = Cd; [x]_u_ = urine concentration of x (mass/volume); and [cr]_u_ = urine creatinine concentration (mg/dL). The ratio [x]_u_/[cr]_u_ was expressed in μg/g of creatinine.

E_x_ was normalized to C_cr_ as E_x_/C_cr_ = [x]_u_[cr]_p_/[cr]_u_, where x = Cd; [x]_u_ = urine concentration of x (mass/volume); [cr]_p_ = plasma creatinine concentration (mg/dL); and [cr]_u_ = urine creatinine concentration (mg/dL). E_x_/C_cr_ was expressed as the excretion of x per volume of filtrate [25].

### 4.5. Statistical Analysis

Data were analyzed with IBM SPSS Statistics 21 (IBM Inc., New York, NY, USA). The one-sample Kolmogorov–Smirnov test was used to identify departures of continuous variables from a normal distribution, and a logarithmic transformation was applied to variables that showed rightward skewing before they were subjected to parametric statistical analysis. The Kruskal–Wallis test was used to assess differences in means among three exposure groups. The chi-square test was used to determine differences in percentage and prevalence data. To determine strength of association of excretion rates of total protein and eGFR with independent variables, the multiple linear regression model analysis was used. The multivariable logistic regression analysis was used to determine the prevalence odds ratio (POR) for albuminuria and CKD in relation to six independent variables: age, gender, diabetes, smoking, hypertension, and Cd exposure, measured as Cd excretion (E_Cd_).

## 5. Conclusions

The conventional method for adjusting excretion rates of Cd and proteins of low and high molecular weight, namely, β_2_M, α_1_M, albumin, and total protein, to creatinine excretion understate the severity of the nephrotoxicity of Cd. The body burden of Cd among residents of an area affected by Cd contamination is indistinguishable when the urinary Cd excretion levels are adjusted to creatinine excretion, thereby nullifying a dose–response relationship analysis. In comparison, normalization of excreted Cd, β_2_M, α_1_M, albumin, and total protein to creatinine clearance enables a dose–effect analysis and provides, for the first time, evidence that proteinuria after chronic Cd exposure is a manifestation of Cd-induced tubulopathy and Cd-induced nephron loss. The results of a dose–effect analysis also provide evidence that albumin is preferentially reabsorbed.

## Figures and Tables

**Figure 1 ijms-24-01893-f001:**
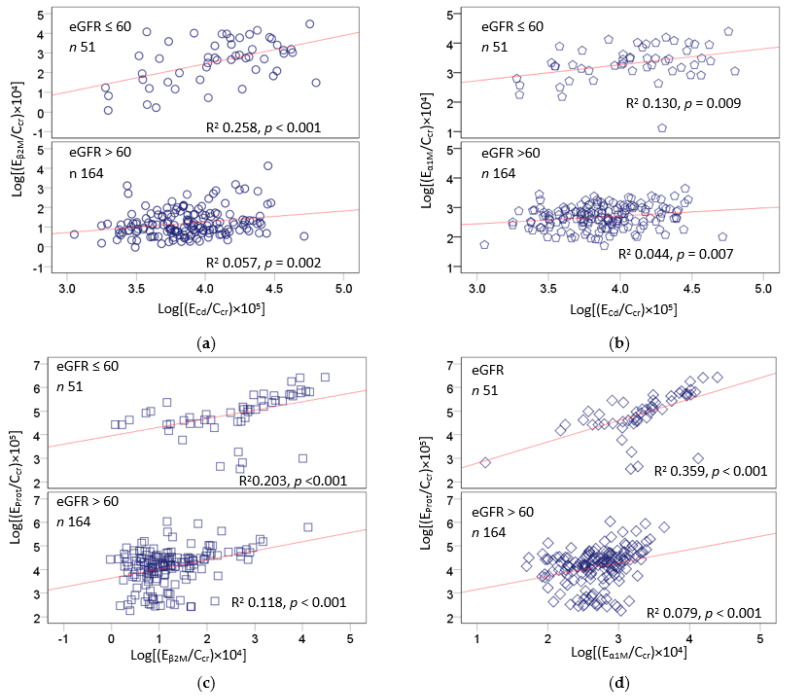
Effects of cadmium exposure and nephron mass on excretion of β_2_-microglobulin and α_1_-microglobulin. Scatterplot (**a**) relates log[(E_βM_/C_cr_) × 10^4^] to log[(E_Cd_/C_cr_) × 10^5^]. Scatterplot (**b**) relates log[(E_α1M_/C_cr_) × 10^4^] to log[(E_C_d/C_cr_) × 10^5^]. Scatterplot (**c**) relates log[(E_Prot_/C_cr_) × 10^5^] to log[(E_β2M_/C_cr_) × 10^4^]. Scatterplot (**d**) relates log[(E_Prot_/C_cr_) × 10^5^] to log[(E_α1M_/C_cr_) × 10^4^] (**d**). Units of E_β2M_/C_cr_, E_α1M_/C_cr_, and E_Prot_/C_cr_ are mg/L of filtrate, and the unit of E_Cd_/C_cr_ is µg/L of filtrate. Coefficients of determination (R^2^) and *p*-values are provided for all scatterplots.

**Figure 2 ijms-24-01893-f002:**
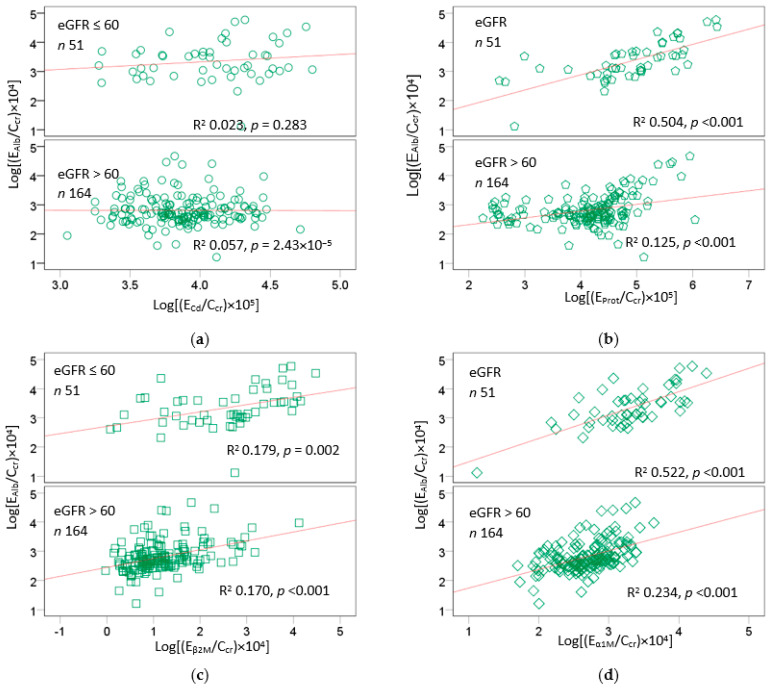
Effects of cadmium exposure and nephron mass on excretion of albumin. Scatterplot (**a**) relates log[(E_Alb_/C_cr_) × 10^4^] to log[(E_Cd_/C_cr_) × 10^5^]. Scatterplot (**b**) relates log[(E_Alb_/C_cr_) × 10^4^] to log[(E_Prot_/C_cr_) × 10^5^]. Scatterplot (**c**) relates log[(E_Alb_/C_cr_) × 10^4^] to log[(E_β2M_/C_cr_) × 10^4^]. Scatterplot (**d**) relates log[(E_Alb_/C_cr_) × 10^4^] to log[(E_α1M_/C_cr_) × 10^4^]. Units of E_Alb_/C_cr_, E_β2M_/C_cr_, and E_α1M_/C_cr_ are mg/L of filtrate, and the unit of E_Cd_/C_cr_ is µg/L of filtrate. Coefficients of determination (R^2^) and *p*-values are provided for all scatterplots.

**Table 1 ijms-24-01893-t001:** Descriptive characteristics of the study subjects according to eGFR levels.

Parameters	All Subjects *n* = 405	eGFR ^a^, mL/min/1.73 m^2^			*p*
		>90, *n* = 207	61–90, *n* = 147	≤60, *n* = 51	
Low-exposure controls (%)	46.9	84.1	10.9	0	<0.001
Females (%)	51.4	48.3	56.5	49.0	0.299
Smoking (%)	45.9	32.9	56.5	68.6	<0.001
Diabetes (%)	2.7	0	4.8	7.8	0.001
Hypertension (%)	13.8	2.9	19.7	41.2	<0.001
Age, years	44.6 ± 16.2	33.2 ± 9.9	53.2 ± 11.5	65.6 ± 10.6	<0.001
eGFR, mL/min/1.73 m^2^	87.3 ± 23.3	106.0 ± 10.1	75.5 ± 8.2	45.4 ± 11.3	<0.001
Plasma creatinine, mg/dL	0.98 ± 0.29	0.84 ± 0.1392	0.98 ± 0.14	1.50 ± 0.44	<0.001
Urine creatinine, mg/dL	106 ± 68	92 ± 68	116 ± 62	135 ± 69	<0.001
Plasma total protein, g/dL	8.04 ± 0.45	8.05 ± 0.45	7.93 ± 0.47	−	0.337
Plasma albumin, g/dL	4.97 ± 0.30	4.97± 0.31	4.98 ± 0.28	−	0.861
Urine protein, mg/L	47.18 ± 150.6	5.42 ± 10.33	50.83 ± 137.3	206.2 ± 307.7	<0.001
Urine protein ≥ 20 mg/L (%)	28.6	3.9	45.6	80.4	<0.001
Urine Cd, µg/L	6.54 ± 10.59	2.24 ± 8.38	9.46 ± 8.22	15.6 ± 15.3	<0.001
Normalized to E_cr_ as E_x_/E_cr_ ^b^					
E_Prot_/E_cr_, mg/g creatinine	43.76 ± 132.57	6.52 ± 11.58	52.36 ± 141.78	170 ± 246	<0.001
E_Prot_/E_cr_ ≥ 100 mg/g creatinine (%)	8.6	0.5	8.8	41.2	<0.001
E_Cd_/E_cr_, µg/g creatinine	5.81 ± 7.64	2.14 ± 5.43	8.95 ± 7.06	11.69 ± 9.20	<0.001
Normalized to C_cr_ as E_x_/C_cr_ ^c^					
(E_Prot_/C_cr_) × 100, mg/L filtrate	60.2 ± 236.5	5.32 ± 9.00	51 ± 134	310 ± 568	<0.001
(E_Prot_/C_cr_) × 100 ≥ 20 mg/L (%)	30.6	3.9	49	86.3	<0.001
(E_Cd_/C_cr_) × 100, µg/L filtrate	6.21 ± 9.00	1.70 ± 4.55	8.67 ± 6.85	17.44 ± 14.17	<0.001

*n*, number of subjects; eGFR, estimated glomerular filtration rate; E_x_, excretion of x; cr, creatinine; C_cr_, creatinine clearance; Prot, protein; Cd, cadmium; ^a^ eGFR, was determined by equations of the Chronic Kidney Disease Epidemiology Collaboration [20]. ^b^ E_x_/E_cr_ = [x]_u_/[cr]_u_; ^c^ E_x_/C_cr_ = [x]_u_[cr]_p_/[cr]_u_, where x = Prot or Cd [25]. Data for all continuous variables are arithmetic means ± standard deviation (SD). Data for plasma protein and plasma albumin are from 190 subjects of the low-exposure control group. Data for all other variables are from all subjects (*n* = 405). For all tests, *p* ≤ 0.05 identifies statistical significance, determined by Pearson chi-square test for % differences and by Kruskal–Wallis test for mean differences across three eGFR subsets.

**Table 2 ijms-24-01893-t002:** Multiple regression analyses to evaluate strength of association of log[(E_Prot_/Ccr × 10^5^) with log log[(E_Cd_/Ccr × 10^5^) and other independent variables.

Independent Variables/ Factors		Urinary Excretion of Protein ^a^				
	All subjects, *n* = 405		Males, *n* = 197		Females, *n* = 208	
	β ^b^	*p*	β	*p*	β	*p*
Age, years	0.263	<0.001	0.222	0.028	0.260	0.011
Log [(E_Cd_/C_cr_) × 10^5^], µg/L filtrate	0.252	<0.001	0.376	<0.001	0.179	0.050
Diabetes	−0.039	0.353	0.012	0.831	−0.097	0.114
Sex	0.078	0.107	−	−	−	−
Hypertension	−0.065	0.152	−0.116	0.069	−0.002	0.974
Smoking	−0.075	0.150	0.007	0.911	−0.152	0.040
Adjusted R^2^	0.306	<0.001	0.371	<0.001	0.259	<0.001

*n*, number of subjects; ^a^ urinary excretion of protein as log[(E_Prot_/C_cr_) ×10^5^]; ^b^ β, standardized regression coefficients. Coding: female = 1, male = 2, hypertension = 1, normotension = 2, smoker = 1, non-smoker = 2. Data were generated from regression model analyses relating E_Prot_ to six independent variables (first column) in all subjects, males, and females. For all tests, *p*-values < 0.05 indicate a statistical significance association. β coefficients indicate the strength of association of E_Prot_ and independent variables. Adjusted R^2^ indicates the proportion of the variation in E_Prot_ attributable to all six independent variables.

**Table 3 ijms-24-01893-t003:** Multiple regression analyses to determine strength of association of eGFR with log[(E_Cd_/Ccr × 10^5^) and other independent variables.

Independent Variables/ Factors		eGFR, mL/min/1.73 m^2 a^				
	All Subjects, *n* = 405		Males, *n* = 197		Females, *n* = 208	
	β ^b^	*p*	β	*p*	β	*p*
Age, years	−0.558	<0.001	−0.603	<0.001	−0.504	<0.001
Log [(E_Cd_/C_cr_) × 10^5^], µg/L filtrate	−0.266	<0.001	−0.178	0.012	−0.334	<0.001
Diabetes	0.034	0.256	0.048	0.246	0.033	0.441
Sex	−0.049	0.155	−	−	−	−
Hypertension	0.084	0.010	0.158	0.001	0.012	0.790
Smoking	−0.043	0.247	−0.061	0.168	0.016	0.750
Adjusted R^2^	0.650	<0.001	0.681	<0.001	0.635	<0.001

*n*, number of subjects; ^a^ eGFR was determined by equations of the Chronic Kidney Disease Epidemiology Collaboration [20]; ^b^ β, standardized regression coefficients. Coding: female = 1, male = 2, hypertension = 1, normotension = 2, smoker = 1, non-smoker = 2. Data were generated from regression model analyses relating eGFR to six independent variables (first column) in all subjects, males, and females. E_Cd_ was as log [(E_Cd_/C_cr_) × 10^5^]. For all tests, *p*-values < 0.05 indicate a statistical significance association. β coefficients indicate strength of association of eGFR and independent variables. Adjusted R^2^ indicates the proportion of the variation in eGFR attributable to all six independent variables.

**Table 4 ijms-24-01893-t004:** Prevalence odds ratios for proteinuria and low eGFR in relation to cadmium excretion and other variables.

Parameters	Number of Subjects	Proteinuria ^a^		Low eGFR ^b^	
		POR (95% CI)	*p*	POR (95% CI)	*p*
Age, years	405	0.923 (0.897, 0.949)	<0.001	0.888 (0.854, 0.924)	<0.001
Diabetes	11	0.726 (0.181, 2.916)	0.652	0.582 (0.119, 2.861)	0.506
Sex (females)	208	1.030 (0.539, 1.971)	0.928	0.775 (0.336, 1.787)	0.550
Hypertension	56	0.498 (0.244, 1.017)	0.055	0.363 (0.159, 0.826)	0.016
Smoking	186	0.778 (0.398, 1.520)	0.462	1.271 (0.523, 3.092)	0.597
E_Cd_/C_cr_ × 100, µg/L filtrate					
0.04–2.71	203	Referent			
2.72–8.28	102	1.252 (0.670, 2.341)	0.482	4.579 (1.116, 18.79)	0.035
8.29–63	100	4.575 (1.880, 11.13)	0.001	5.109 (2.093, 12.47)	<0.001

POR, prevalence odds ratio; CI, confidence interval. Coding: female = 1, male = 2, hypertensive = 1, normotensive = 2, smoker = 1, non-smoker = 2. ^a^ Proteinuria was defined as (E_Prot_/C_cr_) × 100 ≥ 20 mg/L filtrate; ^b^ low eGFR was defined as estimated GFR ≤ 60 mL/min/1.73 m^2^. Data were generated from logistic regression analyses relating POR for proteinuria and low eGFR to a set of six independent variables (first column). For all tests, *p*-values ≤ 0.05 indicate a statistically significant association of POR with a given independent variable. Arithmetic means (SD) of (E_Cd_/C_c_r) × 100 ranges: 0.04–2.71, 2.72–8.28, and 8.29–63 µg/L filtrate were 0.59 (0.54), 5.40 (1.71), and 18.5 (10.5) µg/L filtrate, respectively.

**Table 5 ijms-24-01893-t005:** Comparing excretion rates of various proteins and cadmium in the high-exposure group stratified by eGFR levels.

Parameters	All Subjects *n* = 215	eGFR ^a^, mL/min/1.73 m^2^			*p*
		>90, *n* = 33	61–90, *n* = 131	≤60, *n* = 51	
Age, years	57.0 ± 11.1	49.4 ± 9.4	55.6 ± 9.6	65.6 ± 10.6	<0.001
BMI, kg/m^2^	21.4 ± 3.6	21.2 ± 3.2	21.3 ± 3.5	21.7 ± 4.3	0.822
eGFR, mL/min/1.73 m^2^	71.6 ± 19.4	100.4 ± 8.3	74.6 ± 8.2	45.4 ± 11.3	<0.001
Plasma creatinine, mg/dL	1.07 ± 0.35	0.79 ± 0.13	0.98 ± 0.14	1.50 ± 0.44	<0.001
Urine creatinine, mg/dL	118.4 ± 62.2	99.1 ± 53.1	116.8 ± 60.2	135.2 ± 69.4	0.054
Plasma to urine creatinine ratio	0.0125 ± 0.0096	0.0116 ± 0.0097	0.0118 ± 0.0094	0.0148 ± 0.0098	0.034
Urine Cd, µg/L	11.85 ± 12.28	11.18 ± 18.70	10.56 ± 8.05	15.61 ± 15.31	0.079
Urine β_2_M, mg/L	4.92 ± 17.43	0.20 ± 0.36	1.18 ± 4.02	17.57 ± 32.31	<0.001
Urine α_1_M, mg/L	13.09 ± 18.68	5.66 ± 6.17	8.37 ± 7.91	30.04 ± 30.31	<0.001
Urine albumin, mg/L	25.57 ± 70.59	7.62 ± 7.29	22.64 ± 76.57	44.72 ± 73.74	<0.001
Urine protein, mg/L	85.4 ± 199.1	14.9 ± 22.6	56.2 ± 144.6	206.2 ± 307.7	<0.001
Normalized to E_cr_ as E_x_/E_cr_ ^b^					
E_Cd_/E_cr_, µg/g creatinine	10.43 ± 8.02	10.26 ± 10.35	9.98 ± 6.79	11.69 ± 9.20	0.641
E_β2M_/E_cr_, mg/g creatinine	4.87 ± 16.55	0.23 ± 0.37	1.66 ± 9.72	16.13 ± 27.49	<0.001
E_α1M_/E_cr_, mg/g creatinine	11.34 ± 15.00	5.78 ± 4.95	7.53 ± 6.30	24.72 ± 24.57	<0.001
E_Alb_/E_cr_, mg/g creatinine	23.21 ± 55.07	10.47 ± 15.68	20.71 ± 59.50	37.88 ± 57.23	<0.001
E_Prot_/E_cr_, mg/g creatinine	78.25 ± 174.96	16.73 ± 24.54	57.98 ± 149.26	170.13 ± 246.01	<0.001
E_β2M_/E_cr_, µg/g creatinine (%)					
<100	36.7	51.5	43.5	9.8	<0.001
100–999	37.7	42.4	42.0	23.5	<0.001
1000	25.6	6.1	14.5	66.7	<0.001
E_Alb_/E_cr_, mg/g creatinine (%)					
<30	84.2	93.9	88.5	66.7	<0.001
30–300	15.3	0.1	10.7	33.3	<0.001
>300	0.5	0	0.8	0	0.017
Normalized to C_cr_ as E_x_/C_cr_ ^c^					
(E_Cd_/C_cr_) × 100, µg/L filtrate	11.27 ± 9.89	8.10 ± 9.06	9.67 ± 6.60	17.44 ± 14.17	<0.001
(E_β2M_/C_cr_) × 100, mg/L filtrate	7.74 ± 29.06	0.18 ± 0.28	1.82 ± 11.58	27.82 ± 52.20	<0.001
(E_α1M_/C_cr_) × 100, mg/L filtrate	15.00 ± 28.25	4.46 ± 3.59	7.45 ± 6.63	41.20 ± 48.68	<0.001
(E_Alb_/C_cr_) × 100, mg/L filtrate	29.06 ± 75.93	7.50 ± 9.83	20.23 ± 56.82	65.68 ± 119.75	<0.001
(E_Prot_/C_cr_) × 100, mg/L filtrate	109.9 ± 316.8	13.0 ± 19.1	56.3 ± 141.0	310.2 ± 568.2	<0.001

*n*, number of subjects; eGFR, estimated glomerular filtration rate; E_x_, excretion of x; cr, creatinine; C_cr_, creatinine clearance; Prot, protein; Cd, cadmium; ^a^ eGFR was determined by equations of the Chronic Kidney Disease Epidemiology Collaboration [20]. ^b^ E_x_/E_cr_ = [x]_u_/[cr]_u_; ^c^ E_x_/C_cr_ = [x]_u_[cr]_p_/[cr]_u_, where x = Prot or Cd [25]. Data for all continuous variables are arithmetic means ± standard deviation (SD). For all tests, *p* ≤ 0.05 identifies statistical significance, determined by Kruskal–Wallis test for mean differences across three eGFR ranges.

**Table 6 ijms-24-01893-t006:** The Pearson correlation analysis of chemical compositions of urine samples from 215 residents of a high-exposure area.

Variables	Age	BMI	E_Cd_/C_cr_	E_Prot_/C_cr_	E_Alb_/C_cr_	E_β2M_/C_cr_	E_α1M_/C_cr_
Age							
BMI	−0.248 ***						
E_Cd_/C_cr_	0.778 ***	−0.050					
E_Prot_/C_cr_	0.516 ***	−0.133	0.507 ***				
E_Alb_/C_cr_	0.320 ***	−0.120	0.152 *	0.546 ***			
E_β2M_/C_cr_	0.421 ***	−0.184 **	0.430 ***	0.508 ***	0.537 ***		
E_α1M_/C_cr_	0.368 ***	−0.164 *	0.364 ***	0.508 ***	0.653 ***	0.825 ***	
Creatinine	0.152 **	0.084	0.169 **	0.217 ***	−0.024	−0.067	0.015

Numbers are Pearson’s correlation coefficients. * *p* = 0.016–0.026, ** *p* = 0.001–0.007, *** *p* < 0.001.

## Data Availability

All data are contained within this article.

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
