# Peer review of "Cadmium-Induced Proteinuria: Mechanistic Insights from Dose–Effect Analyses"

_ijms, 2023, doi:10.3390/ijms24031893_

Round 1
Reviewer 1 Report
Comments: The work is an interesting manuscript, However, there are some conceptual errors in this article, Would you please clarify the following arguments before acceptance?
1. Despite the importance of this study, however, the result did not explain the correlation between creatinine and β-microglobulin.
2. the main risk of this manuscript is lacking the histopathological profile of patients associated with chronic kidney disease in their current study and in their previous one (cited in 9).
3. Authors did not explain how this study can be used in the clinic and how can it be useful for physicians and patients.
4. Collagen as a result of kidney fibrosis was not considered in the study, the authors should provide an explanation
Author Response
Reviewer 1
We thank the reviewer for evaluating our manuscript and his/her useful suggestions for improvement. Accordingly, we have substantially revised it taking particular care with the English. Point-by-point responses are given below. Changes to the text are in blue.
Comments and responses.
The work is an interesting manuscript. However, there are some conceptual errors in this article, would you please clarify the following arguments before acceptance?
Point 1: Despite the importance of this study, however, the result did not explain the correlation between creatinine and β2-microglobulin.
Response
We have added urinary creatinine to the correlation analysis in table 6. No correlation between excretion of creatinine and β2-microglobulin was, however, observed. The results of our correlation analysis (lines 226-228) are quoted as below.
“The excretion of creatinine correlated positively with age, ECd/Ccr, and EProt/Ccr. However, this parameter did not correlate with EAlb/Ccr, Eβ2M, or Eα1M/Ccr.”
We suspected that the lack of a positive correlation between excretion of creatinine and β2-microglobulin may be due at least in part to the correlation of creatinine excretion with age which may reflect age related loss of skeletal muscle creatinine. The discrepancy of correlative results due to creatinine excretion variability underscores normalization of excretion rate of a given chemical to clearance of creatinine. We have illustrated in the present study (quoted below) that the body burden of Cd was indistinguishable when ECd was normalized to Ecr (Table 5).
“Relative to the high eGFR group, the excretion of creatinine tended to rise in the moderate and the low eGFR groups (p = 0.054), while the urinary Cd concentrations (µg/L) in three eGFR groups did not differ (p = 0.646) (Table 5). In effect, the body burdens of Cd of subjects in the three eGFR groups were indistinguishable (p = 0.079).”
Point 2: The main risk of this manuscript is lacking the histopathological profile of patients associated with chronic kidney disease in their current study and in their previous one (cited in 9).
Response: Point 2 and point 4 are related to pathogenesis of Cd-induced CKD, they are addressed together below.
Point 3: Authors did not explain how this study can be used in the clinic and how can it be useful for physicians and patients.
Response: We have added in the Discussion below statements.
“The results of the present study have strongly implicated Cd exposure as a risk factor for CKD. Minimization/avoidance of Cd exposure from smoking and habitual consumption of foods containing high levels of the metal are warranted.”
Point 4: Collagen as a result of kidney fibrosis was not considered in the study, the authors should provide an explanation.
Response: A new paragraph has been added to the discussion to at least provide evidence that support Cd-induced tubular atrophy and fibrosis and that Cd increased the mortality from kidney failure (lines 307-314).
“Kidney fibrosis after chronic exposure to Cd has been demonstratable in experimental studies [42, 43]. Evidence from the synchrotron imaging of metals in human kidney tissue samples is in line with Cd-induced fibrosis [44]. The degree of tubular atrophy was positively associated with the level of Cd accumulation in a histopathological examination of kidney biopsies from healthy kidney transplant donors [45]. A prospective cohort study of Japanese residents in an area similarly polluted by Cd reported a 49% increase in mortality from kidney failure among women after adjustment for potential confounding factors [46].”
- Thijssen, S.; Lambrichts, I.; Maringwa, J.; Van Kerkhove, E. Changes in expression of fibrotic markers and histopathological alterations in kidneys of mice chronically exposed to low and high Cd doses. Toxicol. 2007, 238, 200-210.
43.Liang, L.; Huang, K.; Yuan, W.; Liu, L.; Zou, F.; Wang, G. Dysregulations of miR-503-5p and Wnt/β-catenin pathway coor-dinate in mediating cadmium-induced kidney fibrosis. Ecotoxicol. Environ. Saf. 2021, 224, 112667.
44.Barregard L, Sallsten G, Lundh T, Mölne J. Low-level exposure to lead, cadmium and mercury, and histopathological find-ings in kidney biopsies. Environ. Res. 2022, 211, 113119.
45.Gobe, G.C.; Mott, S.A.; de Jonge, M.; Hoy, W.E. Heavy metal imaging in fibrotic human kidney tissue using the synchrotron X-ray fluorescence microprobe. Transl. Androl. Urol. 2019, 8, S184-S191.
46.Nishijo, M.; Nogawa, K.; Suwazono, Y.; Kido, T.; Sakurai, M.; Nakagawa, H. Lifetime cadmium exposure and mortality for renal diseases in residents of the cadmium-polluted Kakehashi River Basin in Japan. Toxics 2020, 8, 81.

Reviewer 2 Report
Dear authors,
1. It is necessary to reformulate the introduction!
2. It is necessary to show the value of total proteins and albumin in the serum as independent variables?
3. Was there a division of patients in the group of diabetics according to KDIGO - defined microalbuminuria?
4. Were there any patients with glomerulopathy or any other recurrent kidney damage that led to a drop in GFR?
5. Are there any results and have any patients ended up temporarily or permanently on some method of replacement of renal function?
Author Response
Reviewer 2
We thank the reviewer for evaluating our manuscript and his/her suggestions for its improvement. Accordingly, we have revised it thoroughly. Point-by-point responses are provided below. Changes to the text are in blue.
Comments and responses
Point 1: It is necessary to reformulate the introduction!
Response:
- The Introduction has been reformulated to reflect the main focus of a study on proteinuria induced by Cd. A paragraph on albumin excretion previously in the Discussion has been placed in the Introduction.
Point 2: It is necessary to show the value of total proteins and albumin in the serum as independent variables?
Response:
- We obtained data on plasma total protein and plasma albumin for the low-exposure control group only, and this data has been added to Table 1.
- As advised, an additional regression analysis has been carried out and results are described in the text (lines 138-143), as below.
“An additional regression analysis of EProt/Ccr was undertaken to assess a potential influence of plasma protein/albumin levels using data from 190 subjects of the low-exposure group. Plasma protein, plasma albumin, age, log [(ECd/Ccr) ×105], sex, and smoking altogether did not explain the EProt/Ccr variability (adjusted R2 = 0.008, p = 0.281). Associations of EProt/Ccr with plasma protein, plasma albumin, and other independent variables were all not significant (p>0.05).”
Point 3: Was there a division of patients in the group of diabetics according to KDIGO - defined microalbuminuria?
Response:
- The KDIGO classification of albuminuria, >30, 30-300 and >300 mg/g creatinine has been inserted to Table 5, and the results of a closer analysis of data according to the Reviewer’s suggestion are provided in the text (lines 213-216) as below.
“Only 1 subject (0.5%) had severely increased albumin excretion (EAlb/Ecr >300 mg/g creatinine). In the majority of subjects (84.2%), EAlb/Ecr values were < 30 mg/g creatinine. Only 15.3% had microalbuminuria (EAlb/Ecr values of 30-300 mg/g creatinine). The % of those with microalbuminuria was lowest, middle and highest in the high-, moderate-, and low eGFR groups, respectively (p < 0.001).
Point 4: Were there any patients with glomerulopathy or any other recurrent kidney damage that led to a drop in GFR?
Response:
- Based on results from an additional analysis of the KDIGO classification of albuminuria, the eGFR reduction seen in Cd-exposed subjects was due mainly to tubulopathy.
- Glomerulopathy was not indicated in the subjects recruited. This may be due to an exclusion criterion of advanced chronic disease.
Point 5: Are there any results and have any patients ended up temporarily or permanently on some method of replacement of renal function?
Response:
- Until now, no official information is available regarding the incidence of kidney failure among residents of a Cd contamination area of Thailand. However, in the discussion, we inferred from a prospective cohort study of Japanese residents in an area similarly polluted by Cd that reported a 49% increase in mortality from kidney failure among women after adjustment for potential confounding factors (Nishijo et al. 2020).
Nishijo, M.; Nogawa, K.; Suwazono, Y.; Kido, T.; Sakurai, M.; Nakagawa, H. Lifetime cadmium exposure and mortality for renal diseases in residents of the cadmium-polluted Kakehashi River Basin in Japan. Toxics 2020, 8, 81.

Reviewer 3 Report
The manuscript titled “Cadmium-Induced Proteinuria: Mechanistic Insights from Dose-Effect Analyses” is aimed to evaluate the concurrent effects of Cd accumulation in kidneys on GFR reduction and excretion rates of proteins of high and low molecular weights, namely total protein, albumin, β2M, and α1M. The topic is relevant since Cd is recognized as the first among 12 dangerous chemical substances of global significance by the United Nations Program, and as the first carcinogen by the International Agency for Research on Cancer. The manuscript is well-designed and deserves attention. However, some issues should be improved before publication.
1. Introduction should be modified. Cd is the most dangerous heavy metal for human health and has various mechanisms of the effect on human organism. Before mentioning and analyzing the effect of cadmium on the kidneys, the authors should briefly mention other negative effects associated with red blood cells damage/anemia (https://doi.org/10.1016/j.ecoenv.2020.111683), oxidative stress (https://doi.org/10.3390/antiox9060492), periodontal disease/bone and cartilage effect (https://doi.org/10.3390/toxics7020031), DNA damage and apoptosis of human liver carcinoma cells (https://doi.org/10.3390/ijerph13010088) etc. Environmental circulation of Cd and mechanism of intake should be mentioned too (soil-plans-animal, drinking water etc).
2. The manuscript should be checked for typos and stylistically aspects. For instance:
Line 165: “0.923, < 0.001 “ – “p” is missed.
Lines 134-135: “…was determined by equations of the Chronic Kidney Disease Epidemiology Collaboration [ ];” is it missed reference?
Line 297. Title of subsection 4.2 is formed differ from other subsections titles
etc.
3. In my opinion, for this work, Materials and method sections should be better the second section, before results. This order will make it easier to first understand the design of the experiment and then study the results. This is just a recommendation.
4. The authors should check that all abbreviations were defined at the first appearance in the text. All units should be defined too. Inverse, CKD was defined twice (L. 49 and L. 63), this should be corrected too.
5. Conclusion should be supported by the most important data obtained.
Generally, I found this manuscript interesting. The authors carried out well statistical justification of presented data. Discussion is supported by references. Abstract reflects the essence of the work. The authors gave details for each equipment/soft. I would recommend this manuscript for publication when the authors will consider and decide all comments.
Author Response
Reviewer 3
We thank the reviewer for evaluating our manuscript and for useful comments and suggestions for improving the introduction and the conclusion. We have revised our paper accordingly. We provide below point-by-point responses to the issues raised by the Reviewer. In a manuscript, changes to the text are in blue.
Comments and responses
The manuscript titled “Cadmium-Induced Proteinuria: Mechanistic Insights from Dose-Effect Analyses” is aimed to evaluate the concurrent effects of Cd accumulation in kidneys on GFR reduction and excretion rates of proteins of high and low molecular weights, namely total protein, albumin, β2M, and α1M. The topic is relevant since Cd is recognized as the first among 12 dangerous chemical substances of global significance by the United Nations Program, and as the first carcinogen by the International Agency for Research on Cancer. The manuscript is well-designed and deserves attention. However, some issues should be improved before publication.
Point 1: Introduction should be modified.
Cd is the most dangerous heavy metal for human health and has various mechanisms of the effect on human organism. Before mentioning and analyzing the effect of cadmium on the kidneys, the authors should briefly mention other negative effects associated with red blood cells damage/anemia (https://doi.org/10.1016/j.ecoenv.2020.111683), oxidative stress (https://doi.org/10.3390/antiox9060492), periodontal disease/bone and cartilage effect (https://doi.org/10.3390/toxics7020031), DNA damage and apoptosis of human liver carcinoma cells (https://doi.org/10.3390/ijerph13010088) etc. Environmental circulation of Cd and mechanism of intake should be mentioned too (soil-plans-animal, drinking water etc).
Response:
- As advised, we have rewritten the introduction to better reflect the focus and content of our paper.
- We have carefully studied the recommended references, and we have elected to cite them where appropriate.
Point 2: The manuscript should be checked for typos and stylistically aspects.
For instance: Line 165: “0.923, < 0.001 “ – “p” is missed.
Response: Corrections have been undertaken.
Lines 134-135: “…was determined by equations of the Chronic Kidney Disease Epidemiology Collaboration [ ];” is it missed reference?
Response: The error has been rectified.
Line 297. Title of subsection 4.2 is formed differ from other subsections titles
etc.
Response: All subtitles in Section 4 have now been uniformly formatted.
Point 3: In my opinion, for this work, materials and method sections should be better the second section, before results. This order will make it easier to first understand the design of the experiment and then study the results. This is just a recommendation.
Response: We have organized sections in accordance to the IJMS mandates.
Point 4: The authors should check that all abbreviations were defined at the first appearance in the text. All units should be defined too. Inverse, CKD was defined twice (L. 49 and L. 63), this
should be corrected too.
Response: CKD and NGAL have now been defined at the first appearance and abbreviations are used thereafter. As for units, common abbreviations for units should not need definition.
Point 5. Conclusion should be supported by the most important data obtained.
Response: The conclusion has been rewritten as suggested.
Point 6: Generally, I found this manuscript interesting. The authors carried out well statistical justification of presented data. Discussion is supported by references. Abstract reflects the essence of the work. The authors gave details for each equipment/soft. I would recommend this manuscript for publication when the authors will consider and decide all comments.
Response: We have carefully revised our manuscript in accordance with the Reviewer’s suggestions and comments.

Round 2
Reviewer 1 Report
The manuscript was revised point by point according to reviewer's comments and It is more acceptable NOW.
Reviewer 2 Report
Nop